# Clinical description and evaluation of 30 pediatric patients with ultra-rare diseases: A multicenter study with real-world data from Saudi Arabia

**Osama Y. Muthaffar**[1], **Noura W. Alazhary**[2], **Anas S. Alyazidi**[3]*, **Mohammed A. Alsubaie**[3‡], **Sarah Y. Bahowarth**[3‡], **Nour B. Odeh**[4‡], **Ahmed K. Bamaga**[1‡]

1 Department of Pediatric, Faculty of Medicine, King Abdulaziz University, Jeddah, Saudi Arabia,
2 Department of General Pediatric, Dr. Soliman Fakeeh Hospital, Jeddah, Saudi Arabia, 3 Faculty of Medicine, King Abdulaziz University, Jeddah, Saudi Arabia, 4 College of Medicine, Alfaisal University, Riyadh, Saudi Arabia

☯ These authors contributed equally to this work.
‡ MAA, SYB, NBO and AKB also contributed equally to this work.
* alyazidi.anas@gmail.com

**Data Availability Statement:** Data cannot be shared publicly because of patients' privacy. Data are available from the Unit of Biomedical Research

## Abstract

### Background

With the advancement of next-generation sequencing, clinicians are now able to detect ultra-rare mutations that are barely encountered by the majority of physicians. Ultra-rare and rare diseases cumulatively acquire a prevalence equivalent to type 2 diabetes with 80% being genetic in origin and more prevalent among high consanguinity communities including Saudi Arabia. The challenge of these diseases is the ability to predict their prevalence and define clear phenotypic features.

### Methods

This is a non-interventional retrospective multicenter study. We included pediatric patients with a pathogenic variant designated as ultra-rare according to the National Institute for Clinical Excellence's criteria. Demographic, clinical, laboratory, and radiological data of all patients were collected and analyzed using multinomial regression models.

### Results

We included 30 patients. Their mean age of diagnosis was 16.77 months (range 3–96 months) and their current age was 8.83 years (range = 2–15 years). Eleven patients were females and 19 were males. The majority were of Arab ethnicity (96.77%). Twelve patients were West-Saudis and 8 patients were South-Saudis. *SCN1A* mutation was reported among 19 patients. Other mutations included *SZT2*, *ROGDI*, *PRF1*, *ATP1A3*, and *SHANK3*. The heterozygous mutation was reported among 67.86%. Twenty-nine patients experienced seizures with GTC being the most frequently reported semiology. The mean response to ASMs was 45.50% (range 0–100%).

at King Abdulaziz University (contact via med. rcommittee@kau.edu.sa) for researchers who meet the criteria for access to confidential data.

**Funding:** The author(s) received no specific funding for this work.

**Competing interests:** The authors have declared that no competing interests exist.

## Conclusion

The results suggest that ultra-rare diseases must be viewed as a distinct category from rare diseases with potential demographic and clinical hallmarks. Additional objective and descriptive criteria to detect such cases are needed.

## Introduction

For millions of children, everyday life is a medical mystery. They battle undiagnosed illnesses, leaving doctors baffled and families desperate. These silent struggles often stem from ultra-rare diseases (URD), debilitating conditions affecting fewer than 1 in 50,000 individuals [1–4]. In recent years, medicine and research underwent huge advancement with the emergence of new diagnostic modalities including next-generation sequencing (NGS) which became the core technology for gene discovery giving practicing clinicians the ability to detect novel mutations [5–8]. NGS can be utilized to scan and sequence thousands of genes including all 22,000 coding genes (a whole exome) or small numbers of individual genes [8] leading to a significant increase in the mutation detection rate [9]. In this sense, many of the newly detected novel variants are rare and barely encountered by the majority of physicians; in the United States, for example, it is estimated that 30 million people suffer from some type of rare disease, a prevalence equivalent to type 2 diabetes [10]. This can lead to underdiagnosing or even misdiagnosing patients with such conditions. Many of those patients subsequently receive symptomatic therapy despite the ability to provide optimal care and avoid patients exposed to unnecessary treatments, unfavorable side effects, and diagnostic odyssey if diagnosed adequately [11]. Furthermore, 80% of rare diseases are genetic in origin [12, 13] and can be inherited which puts additional focus on halting and preventing disease passage to the next generations [14]. The topic is of greater importance in countries such as Saudi Arabia in which the prevalence of consanguineous marriages is estimated to be as high as 50% of all marriages [15]. The increased risk of genetic disorders, foremost rare and URDs, is one of the main concerns associated with consanguineous marriages [16]. Reproduction between closely related individuals increases the likelihood that both parents have the same genetic mutation and subsequently their children are more likely to have recessive genetic disorders as a result of this circumstance [16]. Nonetheless, the diagnostic delay for rare diseases is of broad interval with some literature suggesting a mean of four to five years to reach the correct diagnosis [17–19]. Recently, the term "ultra-rare diseases" was used to describe rare diseases with an incidence of less than one in fifty thousand [1–3] that are likely to exist among patients with undiagnosed rare diseases [20]. The challenges with URD are due to their possibility of representing a very large group of disorders of unknown future prevalence and our ability to define the phenotypic features [21, 22]. In this study, we shed light on seven such URD through the lens of thirty cases encountered at our tertiary care centers. By delving into their clinical features and the hurdles faced in diagnosis and treatment, we aim to contribute to the crucial task of unraveling the mysteries of these hidden illnesses and paving the way for better care for the millions living in uncertainty.

## Materials and methods

### Ethics statement

The study received ethical approval for its protocol and procedures from the Unit of Biomedical Research Ethics at the Faculty of Medicine in King Abdulaziz University on July 12, 2023,

with reference number (265–23). Patients privacy and confidentiality was ensured throughout the conducting of this research and all revealing data were masked accordingly. Written informed consent was obtained to publish the details of the reported cases from the patient's legal parents after describing the nature of the published information, its uses, and the research objectives. This study was performed in accordance with the ethical standards of the Declaration of Helsinki and its later amendments.

## Study design and setting

This is a non-interventional retrospective multicenter study based on an anonymized chart review using the electronic hospital record to identify the targeted population. The study followed the Strengthening the Reporting of Observational Studies in Epidemiology (STROBE) checklist for retrospective studies [23]. The centers of the study included a publicly operated academic tertiary care center, funded and owned, that serves the entire community with a bed capacity of 750. The second center is a privately-owned tertiary care center, with a bed capacity of 300. The third center is a major referral center at the national level mainly providing care to complex cases for patients with special needs with advanced medical services. All three centers are located in a city with high population diversity located on the western coasts of Saudi Arabia. This ensured the heterogeneity of the included sample. Patients aged 14 years and younger currently and following up in the pediatric neurology clinics with a confirmed diagnosis of an URD were included. Diseases identified as ultra-rare according to the following criteria: A) Disease prevalence is 1:50,000 or lower, B) Diseases registered at the National Organization for Rare Disorders (NORD) database for rare diseases. This criteria was adopted in accordance with the National Institute for Clinical Excellence (NICE) definition of URD [24]. The variant classification criteria were based on the American College of Medical Genetics and Genomics (ACMG) recommendations for standards for interpretation and reporting of sequence variations (Class 1: Pathogenic, Class 2: Likely pathogenic, Class 3: Variant of uncertain significance [VUS], Class 4: Likely benign, Class 5: Benign) [25].

## Study procedure

After obtaining ethical clearance, we retrospectively screened the electronic hospital records and included patients visiting the pediatric neurology clinics during the period of January 1 to August 10, 2023. On August 10, 2023, we extracted data directly from the records to minimize errors, focusing on 356 patient visits within that timeframe. This led to the identification of 30 potential cases, whose medical history and demographic information were further reviewed. With informed consent from their families, we then collected detailed data across various categories, including:

- Genetics: genetic variants, homozygosity, alleles, and the specific genetic test used for diagnosis (e.g., whole exome sequencing).

- Demographics: gender, ethnicity, origin, current age, age of diagnosis, type of admission (emergency or outpatient), and baseline laboratory workup

- Birth parameters: term, weight, height, head circumference, pregnancy outcome, neonatal period.

- Seizure characteristics: etiology, semiology, medications used, and response to treatment.

- Clinical information: relevant medical history and observations.

- Developmental history: cognitive, sensorimotor, speech, language, and socioemotional development.

- Imaging and electrophysiological findings: X-rays, MRIs, and electroencephalograms (EEGs).

## Sample collection

Patients with URD were identified using next-generation sequencing (NGS). To implement such procedures, we followed appropriate ethical and logistical measures adopted in our local institution and obtained the genetic sample of the patients at King Abdulaziz University Hospital. DNA capture probes using NGS-based copy number variation (CNV) analysis with Illumina array were performed. The coding regions of the gene and known pathogenic/likely pathogenic variants within the gene (coding and non-coding) were targeted for analysis. Data analysis, variant calling and annotation were performed using validated software [26]. Further prioritization was performed focusing on rare variants that were loss of function (frameshift, nonsense, and splice site mutations), homozygous missense, and/or affecting known disease genes from the Online Mendelian Inheritance in Man database (OMIM). Several prediction tools were used to predict the pathogenicity of the identified variant. The children's variant inheritance mode was compared to the parents' exome sequencing results.

## Data analysis

Data were collected in a Microsoft Excel (Microsoft® Corp., Redmond, WA, USA) version 20 sheet. Statistical analysis was performed using IBM SPSS Statistics for Windows, version 27 (IBM Corp., Armonk, NY, USA) and SmartPLS 3. Numerical variables were analyzed and described as means and standard deviations ($\sigma$). Measures of central tendency were calculated to describe quantitative variables. One-way analysis of variance (ANOVA) and multinomial regression were used to analyze statistical differences between groups. Variable independence was measured by the Likelihood Ratio (LR) Chi statistic and Fisher's exact test. Nominal parameters were compared using the chi-squared test. The Bonferroni correction was used in post hoc testing of the data. The level of significance, ($P$-value), was determined at $<0.05$ and 95% confidence intervals.

## Results

### Patient population

Our study examined a cohort of 30 pediatric patients diagnosed with URD, adhering to established classification criteria. With an average age of 8.83 years (SD 3.90), all participants fell within the pediatric range ($<$ 14 years old). Notably, the average age of diagnosis was significantly later at 16.77 months (SD 19.39). Nineteen participants were male, and the majority (96.77%) identified as Arab ethnicity, with the remaining comprising Pashtun and Berber individuals. Geographical distribution revealed most patients originated from the western and southern regions of Saudi Arabia. Consanguinity within parents was confirmed in a striking 80% of cases, with a reported heterozygous mutation in 67.86% of these familial instances. Among the identified ultra-rare mutations, the *SCN1A* gene (sodium voltage-gated channel alpha subunit 1) stood out, affecting 19 patients. Additional mutations were found in *SZT2* (Seizure Threshold 2), *ROGDI* (Rogdi Atypical Leucine Zipper), *PRF1* (Perforin 1), *ATP1A3* (ATPase Na+/K+ Transporting Subunit Alpha 3), and *SHANK3* (SH3 and multiple ankyrin repeat domains 3) genes across the remaining participants (Fig 1).

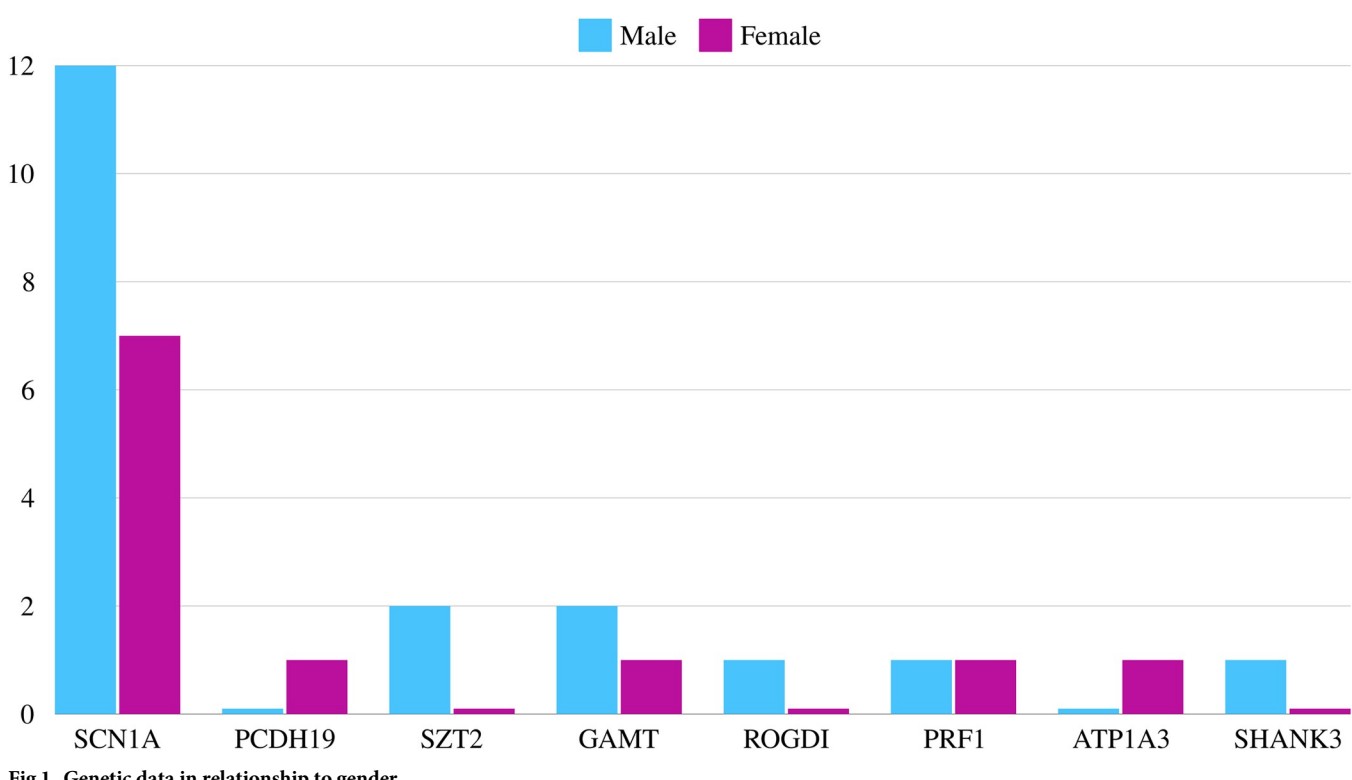

**Fig 1. Genetic data in relationship to gender.**

### Genetic data

Of those mutations, two siblings with a homozygous mutation in *SZT2* c.6113A>Cp. (Tyr2038Ser) (NM_015284.3) suspecting a diagnosis of autosomal recessive developmental and epileptic encephalopathy (DEE) type 18. Their parents were found to be heterozygous for the same mutation in the *SZT2* gene. On the other hand, one patient was found to carry two copies of a specific *ROGDI* gene variant: one inherited from each parent as follows: inherited c.117+1G>A chr16:4802381C>T (NM_024589.3) from this father and c.117+1G>A chr16:4802381C>T (NM_024589.3) from his mother. This variant, classified as likely pathogenic, has been linked to Kohlschutter-Tonz syndrome (KTS). Regarding the patients carrying a variant in the *PRF1* gene, whole exome sequence (WES) testing revealed a homozygous mutation in the *PRF1* gene and was subsequently validated by Sanger sequencing. A sequence variant of c.1081A>T (p.Arg361Trp) (NM_001083116.3) was identified in the patient. Her elder sibling has also revealed a (NM_001083116.3):c.1081A>T (p.Arg361Trp) homozygous variant within the same gene. Both patients were associated with hemophagocytic lymphohistiocytosis (HLH) disease. No parental testing was carried out on the patients. As for the patient carrying a variant in the *ATP1A3* gene, her diagnosis was established using whole genome sequencing (WGS) after performing an inconclusive WES testing. Her mutation was a heterozygous, potential de novo, VUS of (ATP1A3, c.29C>T, p-[Ser10Leu]). Also, two homozygous variants were identified in this patient as follows, (NHLRC2, c.335G>T, p. (Gly112Val) and MAPRE2 c.55C>T, p.[GIn19*]) which do not match with clinical phenotypes. The second patient carrying a variant in the same gene with a potential de novo mutation and heterozygous genotype of *ATP1A3*, c.2366C>A, p.(Pro789Gln) which is likely a pathogenic variant. Similarly to the first case, *ATP1A3* mutation was detected by WGS rather than WES. Concerning the patient carrying the *SHANK3* variant, his WES identified a heterozygous likely

Table 1. Demographics and clinical characteristics of patients with URD.

| Characteristic | Number (total = 30) | % |
|---|---|---|
| **Gender** | | |
| Male | 19 | 63.3 |
| Female | 11 | 36.7 |
| **Origin** | | |
| West Saudi | 12 | 40 |
| South Saudi | 8 | 26.7 |
| North Saudi | 5 | 16.7 |
| Central Saudi | 2 | 6.7 |
| East Saudi | 1 | 3.3 |
| South-east Saudi | 1 | 3.3 |
| Pakistani | 1 | 3.3 |
| **Ethnicity** | | |
| Arab | 28 | 93.3 |
| Arab-Berber | 1 | 3.3 |
| Pashtuns | 1 | 3.3 |
| **Age in years, mean ± SD** | | |
| 8.83 ± 3.90 | | |
| **Age group** | | |
| Toddler | 2 | 6.7 |
| Preschool | 7 | 23.3 |
| School age | 16 | 53.3 |
| Adolescent | 5 | 16.7 |
| **Consanguinity** | | |
| Positive | 24 | 80 |
| Negative | 6 | 20 |
| **Variant** | | |
| *SCN1A* | 19 | 63.3 |
| GAMT | 3 | 10.0 |
| *PRF1* | 2 | 6.7 |
| *SZT2* | 2 | 6.7 |
| *PCDH19* | 1 | 3.3 |
| *ROGDI* | 1 | 3.3 |
| *ATP1A3* | 1 | 3.3 |
| *SHANK3* | 1 | 3.3 |

pathogenic variant in the *SHANK3* gene, (Chr22(GRCh37):g.51153476G>A, NM_001080420.1:c.2313+1G>A), the c.2313+1G>A variant is predicted to disrupt the highly conserved donor splice site of exon 20. A genetic diagnosis of autosomal dominant Phelan-McDermid syndrome was confirmed in this patient. Table 1 shows detailed patients' demographics and clinical characteristics.

## Clinical and electrographic spectrum

Table 2 comprehensively outlines the clinical and behavioral features, developmental delays, comorbidities, as well as magnetic resonance imaging (MRI) and electroencephalogram (EEG) abnormalities in patients affected by URD. In the examined patient cohort, 20% displayed abnormal gait, and 13.3% presented with infantile hypotonia. Attention-deficit/hyperactivity

**Table 2. Clinical and behavioral features, developmental delays, comorbidities, MRI, and EEG abnormalities in patients with URD.**

| Feature | Number (total = 30) | % |
|---|---|---|
| Head and face abnormalities | | |
| Macrocephaly | 2 | 6.7 |
| Amelogenesis imperfecta | 1 | 3.3 |
| Prominent ears | 1 | 3.3 |
| Eyes abnormalities | | |
| Strabismus | 2 | 6.7 |
| Visual impairment | 1 | 3.3 |
| Motor and tone abnormalities | | |
| Infantile hypotonia | 4 | 13.3 |
| Abnormal gait | 6 | 20 |
| Ataxia | 2 | 6.7 |
| Behavioral abnormalities | | |
| ASD | 3 | 10 |
| ADHD | 7 | 23.3 |
| Impulsivity | 2 | 6.7 |
| Aggressiveness | 5 | 16.7 |
| Memory problems | 4 | 13.3 |
| Developmental delay | | |
| GDD | 8 | 26.7 |
| Cognitive delay | 11 | 36.7 |
| Social delay | 9 | 30 |
| Gross motor delay | 6 | 20 |
| Fine motor delay | 8 | 26.7 |
| Speech delay | 14 | 46.7 |
| Comorbidities | | |
| Recurrent infections | 5 | 16.7 |
| Failure to thrive | 2 | 6.7 |
| Leukoencephalopathy | 2 | 6.7 |
| Viral encephalopathy | 1 | 3.3 |
| EEG and MRI abnormalities | | |
| Abnormal EEG | 16 | 53.3 |
| Abnormal MRI | 5 | 16.7 |

**Abbreviations:** ASD: Autism spectrum disorder; ADHD: Attention-deficit/hyperactivity disorder; GDD: Global developmental delay; EEG: electroencephalogram; MRI: Magnetic resonance imaging.

disorder (ADHD) was noted in 23.3% of the patients, while 10% were diagnosed with autism spectrum disorder. Global developmental delay was evident in 26.7% of cases, with speech impairment being the most prevalent manifestation (46.7%). Abnormal EEG findings were documented in 53.3% of patients, and 16.7% exhibited abnormalities in MRI scans.

## Seizures and antiseizure medications

Seizures were observed in 28 cases with generalized tonic-clonic being the most reported (76.7%), followed by febrile seizures (40%) and myoclonic seizures (23.3%). **Fig 2** illustrates the distribution of observed seizure types. **Table 3** shows the antiseizure medications (ASMs) used for patients with URD, outlining both current usage and previous unsuccessful trials.

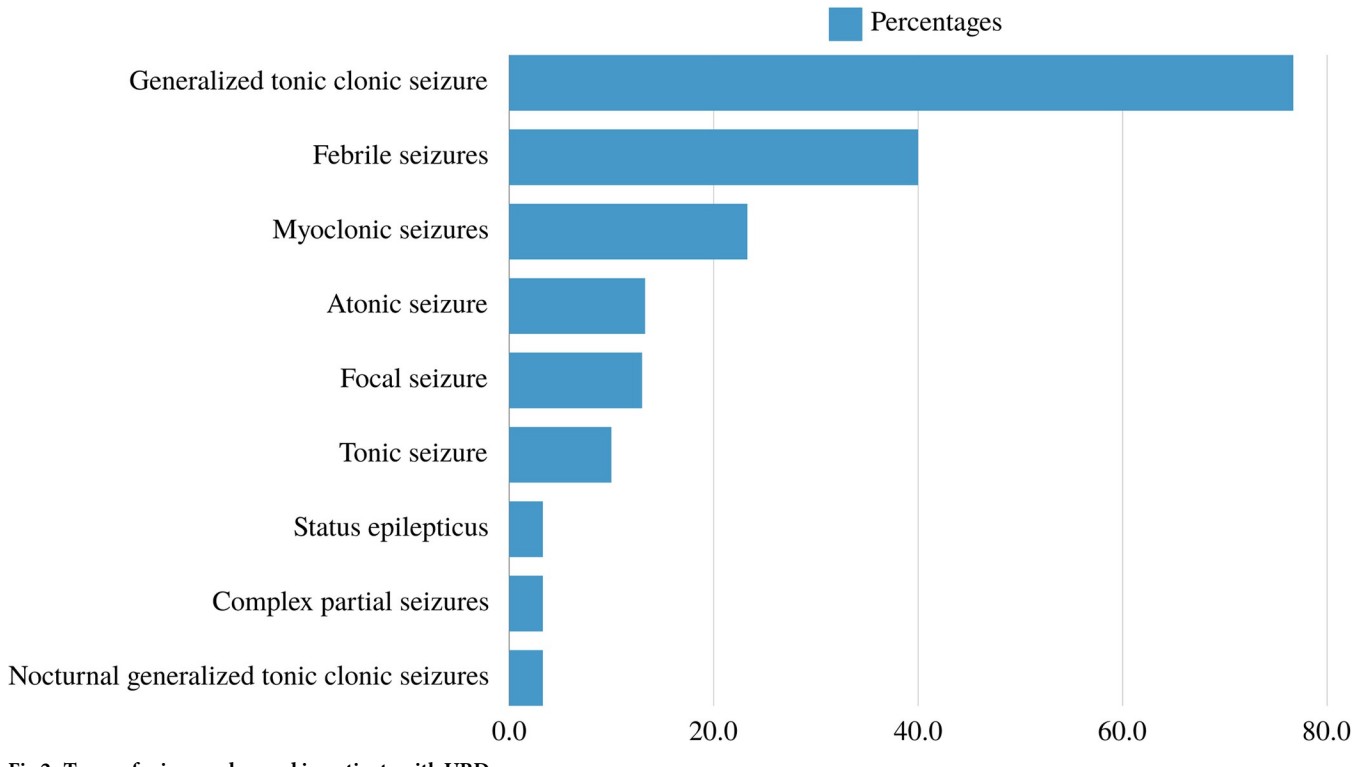

**Fig 2. Types of seizures observed in patients with URD.**

**Table 3. ASM for patients with URD (current use and past failed trials).**

| Administered ASM | Current use, N (%) | Past failed trials, N (%) |
|---|---|---|
| Levetiracetam | 12 (40) | 10 (33.3) |
| Clobazam | 16 (53.3) | 3 (10) |
| Valproic acid | 18 (60) | 7 (23.3) |
| Perampanel | 5 (16.7) | 2 (6.7) |
| Lacosamide | 3 (10) | 0 (0.0) |
| Topiramate | 9 (30) | 8 (26.7) |
| Stiripentol | 4 (13.3) | 0 (0.0) |
| Clonazepam | 1 (3.3) | 5 (16.7) |
| Diazepam | 2 (6.7) | 0 (0.0) |
| Phenobarbitone | 2 (6.7) | 3 (10) |
| Lamotrigine | 2 (6.7) | 7 (23.3) |
| Carbamazepine | 2 (6.7) | 7 (23.3) |
| Buccal Midazolam | 1 (3.3) | 0 (0.0) |
| Steroids | 0 (0.0) | 1 (3.3) |
| Phenytoin | 0 (0.0) | 1 (3.3) |
| Oxcarbazepine | 0 (0.0) | 1 (3.3) |
| Rufinamide | 0 (0.0) | 3 (10) |
| Vigabatrin | 0 (0.0) | 1 (3.3) |
| Ethosuximide | 0 (0.0) | 1 (3.3) |
| Cannabis | 0 (0.0) | 1 (3.3) |

**Abbreviation:** ASM: Antiseizure medication.

Table 4. Response to ASM according to the variant.

| Variant | Number | Mean response | Standard deviation | 95% Confidence Interval for Mean | |
|---|---|---|---|---|---|
| | | | | Lower Bound | Upper Bound |
| SCN1A | 19 | 36.84 | 31.412 | 21.70 | 51.98 |
| PCDH19 | 1 | 25.00 | 0 | 0 | 0 |
| SZT2 | 2 | 90.00 | 14.142 | -37.06 | 217.06 |
| GAMT | 3 | 66.67 | 28.868 | -5.04 | 138.38 |
| ROGDI | 1 | 100.00 | 0 | 0 | 0 |
| PRF1 | 2 | 0 | 0 | .00 | .00 |
| ATP1A3 | 1 | 80.00 | 0 | 0 | 0 |
| SHANK3 | 1 | 80.00 | 0 | 0 | 0 |
| Total | 30 | 45.50 | 35.364 | 32.29 | 58.71 |

**Abbreviation:** ASM: Antiseizure medication.

Valproic acid (60%), clobazam (53.3%), and levetiracetam (40%) were the most currently used ASMs. However, levetiracetam had the highest failure rate (33.3%). Most of the patients (73.3%) were currently on three or more ASMs, with an overall mean response of 45.50%. The most notable response, reaching 100%, was observed in a patient with the *ROGDI* variant. Following closely were two patients with the *SZT2* variant, exhibiting a response rate of 90%. Both *ATP1A3* and *SHANK3* mutations showed a response rate of 80%. Patients with GAMT, *SCN1A*, and *PCDH19* mutations demonstrated response rates of 66.67%, 36.84%, and 25%, respectively. Conversely, patients with the *PRF1* variant displayed no response to ASMs. The detailed breakdown of the response to ASM for each genetic mutation is presented in **Table 4**.

## Discussion

Pediatric URD, defined by their exceptional rarity and devastating impact, presents a crucial but daunting research frontier. Their low prevalence fuels a cascade of challenges: limited understanding, scarce data, and frequent diagnostic delays or misdiagnoses [27]. This "diagnostic odyssey" inflicts emotional and financial burdens on families and emphasizes the urgent need for heightened awareness among healthcare providers, advanced diagnostic tools for faster and more accurate diagnoses, and collaborative efforts to expedite interventions and improve patient outcomes [28]. Beyond clinical care, research on these enigmatic conditions faces its own formidable obstacles. Limited patient pools, fragmented data, and the absence of standardized protocols obstruct the design and execution of clinical trials and studies. To overcome these hurdles, innovative approaches are key, including international collaborations to pool resources and expertise, establishment of patient registries to facilitate data collection and research efforts, and leveraging cutting-edge technologies like genomics to unlock disease mechanisms and identify potential therapeutic targets [29].

Our study included thirty patients with URD according to the previously established classification (**Table 5**). The average age at the time of data collection was 8.83 years and the mean age of diagnosis was found to be 16.77 months. All our studied patients were within the pediatric age group (<14 years old). Most of them were of Arab ethnicity (96.77%) others included Pashtun and Berber. Consanguinity among parents existed in most cases (80%). Furthermore, our patient population showed positive results for the *SCN1A (sodium voltage-gated channel alpha subunit 1)* mutation which was detected in 19 patients. Additionally, we identified mutations in *SZT2 (Seizure Threshold 2)*, *ROGDI (Rogdi Atypical Leucine Zipper)*, *PRF1 (Perforin 1)*, *ATP1A3 (ATPase Na+/K+ Transporting Subunit Alpha 3)*, and *SHANK3 (SH3 and multiple*

**Table 5. Comprehensive overview of characteristics in URD cases.**

| Patient No. | Sex / Years | Region | Genetic Mutations | Consanguinity | Age of diagnosis | Seizure type | ASM (current) | Response rate |
|---|---|---|---|---|---|---|---|---|
| 1 | M / 13 | Southern | SCN1A / Heterozygous | Positive | 6m | Focal febrile seizure, GTC seizure | Valproic acid, diazepam, topiramate, | 50% |
| 2 | F / 14 | Southern | SCN1A / Heterozygous | Positive | 6m | Focal febrile seizure, GTC seizure | Levetiracetam, perampanel, clobazam, valproic acid | 50% |
| 3 | F / 6 | Southern | SCN1A / Heterozygous | Positive | 10m | Focal febrile seizure, GTC seizure | Stripintol, phenobarbitone, clobazam | 0% |
| 4 | M / 12 | Central | SCN1A / Heterozygous | Negative | 6m | Focal febrile seizure, GTC seizure, Myoclonic seizure | Levetiracetam, valproic acid, Topiramate | 0% |
| 5 | M / 6 | Southern | SCN1A / Heterozygous | Positive | 4m | Focal febrile seizure, GTC seizure | Perampanel, clobazam, levetiracetam, Topiramate | 100% |
| 6 | F / 10 | Northern | SCN1A / Heterozygous | Positive | 8m | Focal febrile seizure, GTC seizure | Valproic acid, clobazam | 0% |
| 7 | M / 9 | Southern | SCN1A / Heterozygous | Positive | 6m | Complex partial seizure, GTC seizure, Myoclonic seizure, Atonic seizure | Clobazam, stripintol, Valproic acid | 0% |
| 8 | M / 11 | Western | SCN1A | Negative | 4m | GTC seizure, Tonic seizure | Lacosamide, stripintol, diazepam | 30% |
| 9 | F / 10 | Southern | SCN1A / Heterozygous | Positive | 1y | GTC seizure, Focal febrile seizure | Levetiracetam, clobazam, Topiramate | 50% |
| 10 | M / 4 | Northern | SCN1A / Heterozygous | Positive | 4m | Focal Febrile seizure, GTC seizure, Myoclonic seizure | Perampanel, levetiracetam, valproic acid, lamotrigine | 60% |
| 11 | M / 15 | Western | SCN1A / Heterozygous | Positive | 1y | Focal seizure, GTC seizure | Lacosamide, lamotrigine, clobazam | 0% |
| 12 | M / 6 | Eastern | SCN1A | Positive | 6m | Atonic seizure, GTC seizure, Myoclonic seizure | Perampanel, valproic acid, phenobarbitone | 50% |
| 13 | F / 5 | Western | SCN1A / Heterozygous | Negative | 6m | GTC seizure | Clobazam | 60% |
| 14 | F / 8 | Western | SCN1A / Heterozygous | Positive | 1y | GTC seizure, Focal febrile seizure | Clobazam, Levetiracetam, valproic acid | 25% |
| 15 | F / 15 | Southern | SCN1A | Positive | 4m | GTC seizure | Clobazam, Levetiracetam, valproic acid | 25% |
| 16 | F / 15 | Western | PCDH19 / Heterozygous | Positive | 1y | GTC seizure, Focal seizure | topiramate, valproic acid, levetiracetam | 25% |
| 17 | M / 12 | Western | SCN1A / Heterozygous | Positive | 6m | GTC seizure, Focal febrile seizure | Clobazam, valproic acid, stripintol | 0% |
| 18 | M / 6 | Western | SCN1A / Heterozygous | Negative | 7m | GTC, Focal seizure | clonazepam, valproic acid, Topiramate | 50% |
| 19 | M / 12 | Northern | SZT2 / Homozygous | Positive | 2y | GTC seizure | valproic acid, Carbamazepine, buccal midazolam | 100% |
| 20 | M / 2 | Northern | SZT2 / Homozygous | Positive | 2y | GTC seizure | Levetiracetam, Carbamazepine, | 80% |
| 21 | M / 9 | Western | GAMT / Homozygous | Positive | 2y | GTC seizure, Myoclonic seizure | valproic acid | 50% |
| 22 | M / 4 | Central | GAMT / Homozygous | Positive | 2y | Atonic seizure, Tonic seizure | Clobazam | 100% |
| 23 | F / 8 | Western | GAMT / Homozygous | Positive | 2y | Myoclonic seizure, Tonic seizure | Valproic acid, clobazam | 50% |
| 24 | M / 12 | Northern | ROGDI / Homozygous | Positive | 3m | GTC seizure, Focal febrile seizure | Levetiracetam | 100% |
| 25 | F / 5 | Western | PRF1 / Homozygous | Positive | 5y | - | - | - |
| 26 | M / 11 | Western | PRF1 / Homozygous | Positive | 8y | - | - | - |

*(Continued)*

**Table 5.** (Continued)

| Patient No. | Sex / Years | Region | Genetic Mutations | Consanguinity | Age of diagnosis | Seizure type | ASM (current) | Response rate |
|---|---|---|---|---|---|---|---|---|
| 27 | F / 2 | Western | ATP1A3 / Heterozygous | Positive | 9m | Focal seizure, Focal febrile seizure | Levetiracetam, clobazam, Topiramate, Lacosamide | 80% |
| 28 | M / 11 | Pakistani | SHANK3 / Heterozygous | Negative | 2y | Myoclonic seizure, and Atonic seizure | Levetiracetam, Valproic acid, Topiramate | 80% |
| 29 | M / 4 | Southern | SCN1A / Heterozygous | Negative | 3y | Nocturnal GTC, Status Epilepticus | Valproic acid, Topiramate, clobazam, Perampanel | 60% |
| 30 | M / 8 | South-east Saudi | SCN1A / Heterozygous | Positive | 2y | Focal febrile seizure, GTC seizure | Valproic acid, clobazam | 90% |

**Abbreviation:** ASM: Antiseizure medication.

*ankyrin repeat domains 3)* genes among the remaining patients. These genes have been rarely reported in previously published studies in Saudi Arabia.

Molecularly, the *SCN1A* mutation, which was detected in 19 patients, encodes the alpha subunit of the sodium voltage-gated channel, which plays a crucial role in the generation and propagation of action potentials in neurons [30, 31]. Mutations in *SCN1A* have been associated with various seizure disorders and epilepsy syndromes [32]. The identified *SCN1A* mutations may have significant implications for protein function. These mutations can disrupt the normal functioning of the sodium channel, leading to altered excitability and impaired neuronal signaling [31]. This can result in increased susceptibility to seizures and contribute to the pathogenesis of epilepsy in affected individuals [31, 32]. Furthermore, we identified mutations in *SZT2*, which is involved in regulating seizure threshold and synaptic transmission [33]. The exact role of *SZT2* in epilepsy is still under investigation, but mutations in this gene have been implicated in various forms of epilepsy [34]. The interpretation of *SZT2* mutations requires careful examination of their potential impact on protein function. Disruptive mutations in *SZT2* may affect its ability to modulate synaptic activity and neuronal excitability, leading to an increased risk of seizures [35]. Moreover, as for the *ROGDI* mutation, it may affect protein stability, interactions with other cellular components, or signaling pathways involved in neuronal development and function and eventually causing KTS [36]. Also, the interpretation of *PRF1* mutation involves assessing their impact on perforin function and immune response regulation [37, 38]. The mutation can impair perforin-mediated cytotoxicity, leading to defective immune surveillance and hyperinflammatory responses [39]. Additionally, an *ATP1A3* variant have been associated with a spectrum of neurological disorders which includes alternating hemiplegia of childhood (AHC) and rapid-onset dystonia-parkinsonism (RDP) [40]. Interpreting *ATP1A3* mutations involves assessing their impact on ion transport and neuronal excitability [41]. Mutations can disrupt the function of the Na+/K+-ATPase pump, leading to altered ion homeostasis and impaired neuronal signaling [41]. These disruptions can manifest as neurological symptoms, such as hemiplegia, dystonia, and parkinsonism [42]. Moving forward to *SHANK3* which is a postsynaptic scaffolding protein involved in the organization and function of synapses [43], a mutation in *SHANK3* have been primarily associated with Phelan-McDermid syndrome, similarly in our patient, the syndrome is characterized with a neurodevelopmental disorder characterized by intellectual disability, autism spectrum disorder, and seizures [44]. The sequel is due to a structure and function of the SHANK3 protein, leading to synaptic dysfunction and altered neuronal circuits [43]. Disease severity, age of onset, disease progression, and comorbidities may also be important factors impacting the effectiveness of ASMs.

Clinically, seizures can be a common manifestation of various URD affecting children [45]. In our study, Seizures were observed in 28 cases with generalized tonic-clonic being the most reported (76.7%), followed by febrile seizures (40%) and myoclonic seizures (23.3%). These seizures often present unique challenges due to the rarity and complexity of the underlying conditions. Many URD, such as certain metabolic disorders, neuronal migration disorders, or specific genetic syndromes, may have seizures as a prominent feature [45, 46]. The occurrence of seizures in these conditions can vary widely in presentation and severity, making diagnosis and management intricate tasks [45]. Moreover, seizures presenting with URD might be refractory to standard antiepileptic medications, necessitating tailored treatment approaches that address the specific genetic or metabolic abnormalities driving the seizures [45]. Most of our patients (73.3%) were on three or more ASMs, with an overall mean response of 45.50%. Comprehensive care involving multidisciplinary teams comprising neurologists, geneticists, and other specialists is crucial for accurate diagnosis, ongoing monitoring, and implementing personalized treatment strategies to manage seizures effectively in children with URD [47]. Regarding seizure response, patients reported variation in their response which can be attributed to multiple causes. This can include the gene-drug interactions, polymorphisms, or specific genetic variants that could influence drug metabolism, target engagement, or drug response pathways [48]. Also, factors such as drug absorption, distribution, metabolism, and elimination, as well as target engagement and downstream signaling pathways, can influence the efficacy of ASMs [49]. Recently, evidence of genetic markers, neuroimaging findings, EEG findings, or other molecular indicators, can serve as predictors of treatment response [50, 51].

Moreover, continued research endeavors aim to uncover the underlying mechanisms of seizures in these conditions, paving the way for more targeted and effective therapies that improve the quality of life for affected children. Nonetheless, numerous research studies have tackled the intricate landscape of URD within pediatric populations, striving to illuminate the challenges and potential pathways for better understanding and managing these conditions. A study published in the Orphanet Journal of Rare Diseases (2023) delved into the epidemiology and clinical characteristics of URD in children [52]. Through conducting interviews with patients and their families, the study emphasized the substantial diagnostic delays and the profound psychological impact on patients and families [52]. Also, another article published in 2020 discussed the emerging role of genomic technologies, such as whole-exome sequencing and gene panel testing, in elucidating the genetic basis of URD in pediatric cohorts, emphasizing the potential for personalized and targeted therapies. These studies collectively underscore the complexity of these conditions, emphasizing the imperative for collaborative efforts, innovative diagnostic tools, and targeted interventions to address the unmet needs of children affected by URD [53]. Fueled by its high consanguinity rate and potential link to URD, Saudi Arabia is emerging as a critical hub for research in this challenging field. By illuminating the unique epidemiological, diagnostic, and clinical aspects within the Saudi context, researchers can shed light on these enigmatic conditions and tailor solutions for the region's specific needs. Several landmark studies published in medical journals have paved the way for this growing area of interest. One recent example, published in 2023, explored the diverse spectrum of ultra-rare genetic disorders affecting children in the Saudi healthcare system. This study highlighted the challenges in managing these conditions and emphasized the need for enhanced diagnostic capabilities and multidisciplinary care approaches [54]. Another study, published in 2017, focused on specific URD, revealing crucial insights into the unique genetic landscape and disease patterns prevalent in Saudi children [55]. Our study adds immense value to the current literature as URD is not studied sufficiently. This calls for more efforts towards early genetic testing and early diagnosis to help establish appropriate care and increase the patient's life expectancy.

## Limitations

The study was limited by the absence of comparative literature to compare the findings and the inability to perform functional analysis due to resource constraints. This limitation arises when studying URD that have limited existing literature especially for patients with similar ethnicity and background.

## Conclusion

The research indicates that classifying URD separately from rare diseases may be warranted due to potential differences in demographics and clinical presentation. Establishing additional objective and descriptive criteria for case identification is crucial. Additionally, our findings emphasize the importance of early genetic testing and diagnosis for individuals with URD. Timely identification of these mutations enables physicians to offer accurate information, appropriate counseling, and tailored care plans to affected patients and their families. Early diagnosis also facilitates the implementation of targeted interventions and personalized therapies, which can significantly improve patient outcomes and quality of life. Furthermore, our study underscores the need for enhanced diagnostic capabilities and multidisciplinary care approaches. The complexity and diversity of URD necessitate collaborative efforts between various medical specialties. By leveraging innovative diagnostic tools, such as gene testing, we can enhance our understanding of these conditions and optimize patient management strategies.

## Supporting information

**S1 Checklist. Human participants research checklist.** https://figshare.com/s/ 39d3505fb878d39365b9.
(DOCX)

## Acknowledgments

The authors express deep gratitude to all families and staff of patients.

## Author Contributions

**Conceptualization:** Osama Y. Muthaffar, Noura W. Alazhary, Anas S. Alyazidi.

**Data curation:** Osama Y. Muthaffar, Anas S. Alyazidi, Sarah Y. Bahowarth.

**Formal analysis:** Osama Y. Muthaffar, Noura W. Alazhary, Mohammed A. Alsubaie.

**Investigation:** Osama Y. Muthaffar, Mohammed A. Alsubaie, Sarah Y. Bahowarth, Ahmed K. Bamaga.

**Methodology:** Osama Y. Muthaffar, Noura W. Alazhary, Mohammed A. Alsubaie.

**Project administration:** Osama Y. Muthaffar, Noura W. Alazhary.

**Software:** Mohammed A. Alsubaie, Sarah Y. Bahowarth.

**Supervision:** Osama Y. Muthaffar, Noura W. Alazhary, Ahmed K. Bamaga.

**Validation:** Osama Y. Muthaffar, Noura W. Alazhary, Mohammed A. Alsubaie.

**Visualization:** Sarah Y. Bahowarth, Nour B. Odeh.

**Writing – original draft:** Osama Y. Muthaffar, Noura W. Alazhary, Anas S. Alyazidi, Mohammed A. Alsubaie, Sarah Y. Bahowarth, Nour B. Odeh, Ahmed K. Bamaga.

**Writing – review & editing:** Osama Y. Muthaffar, Noura W. Alazhary, Anas S. Alyazidi, Mohammed A. Alsubaie, Sarah Y. Bahowarth, Nour B. Odeh, Ahmed K. Bamaga.

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
