## [Decision Letter · Decision Letter 0]

4 Apr 2024

PONE-D-24-05215Clinical Description and Evaluation of 30 Pediatric Patients with Ultra Rare Diseases: A Multicenter Study with Real-World Data from Saudi ArabiaPLOS ONE

Dear Dr. Alyazidi,

Thank you for submitting your manuscript to PLOS ONE. After careful consideration, we feel that it has merit but does not fully meet PLOS ONE’s publication criteria as it currently stands. Therefore, we invite you to submit a revised version of the manuscript that addresses the points raised during the review process.

We look forward to receiving your revised manuscript.

Kind regards,

Laith N. Al-Eitan

Academic Editor

PLOS ONE

Journal Requirements:

2. For studies involving third-party data, we encourage authors to share any data specific to their analyses that they can legally distribute. PLOS recognizes, however, that authors may be using third-party data they do not have the rights to share. When third-party data cannot be publicly shared, authors must provide all information necessary for interested researchers to apply to gain access to the data. (https://journals.plos.org/plosone/s/data-availability#loc-acceptable-data-access-restrictions) 

Reviewers' comments:

Reviewer's Responses to Questions

**Comments to the Author**

1. Is the manuscript technically sound, and do the data support the conclusions?

Reviewer #1: Yes

Reviewer #2: Partly

2. Has the statistical analysis been performed appropriately and rigorously? 

Reviewer #1: Yes

Reviewer #2: Yes

3. Have the authors made all data underlying the findings in their manuscript fully available?

Reviewer #1: Yes

Reviewer #2: Yes

4. Is the manuscript presented in an intelligible fashion and written in standard English?

Reviewer #1: Yes

Reviewer #2: Yes

5. Review Comments to the Author

Reviewer #1: Muthaffar et al. used advancement of next-generation sequencing to detect ultra-rare mutations that are barely encountered by the majority of physicians. Ultra-rare and rare diseases cumulatively acquire a prevalence equivalent to type 2 diabetes with 80% being genetic in origin and more prevalent among high consanguinity communities including Saudi Arabia. SCN1A mutation was reported among 19 patients out of total 30 patients. Other mutations included SZT2, ROGDI, PRF1, ATP1A3, and SHANK3. The results suggest that ultra-rare diseases must be viewed as a distinct category from rare diseases with potential demographic and clinical hallmarks. Additional objective and descriptive criteria to detect such cases are needed.

This is an excellent study that cover new aspect of next generation sequencing technique to detect ulta rare mutations in the population. I have some comments before the acceptance of this manuscript for the publications.

Comments:

1. Introduction of the study is very brief need more details with latest reference from the literature is required.

2. It would be helpful to provide more specific information on the prevalence of reported mutations in the Saudi population and the impact of consanguineous marriages on the occurrence of genetic diseases.

3. Please provide the ethical approval of the study.

4. Also, add the details either the consent from patients were taken for this study?

5. It is not clear how many samples were used for WGS and how many samples for WES.

6. It would be better to add graphical representation of the data as done in figure 1. Add more figures that will add value to the manuscript.

7. Detailed methodology is required such as WES and WGS etc.

8. The results section presents the identified variant in the epileptic genes. However, it would be beneficial to include the frequency or prevalence of this variant in the Saudi population, if available. This information would help assess the rarity or novelty of the variant and its potential contribution to the reported clinical presentation.

9. What was inclusion and exclusion criteria of the patients added in this study.

10. The conclusion summarizes the main findings of the study. However, it would be helpful to reiterate the clinical implications of the identified genes and highlight its relevance for genetic counseling and patient management. This would emphasize the practical implications of the research.

11. Some grammatical mistakes and missing reference are seen need to be corrected in the revised manuscript.

Reviewer #2: Present retrospective multicenter study focused on pediatric patients with ultra-rare mutations, revealing their prevalence and clinical characteristics. Results highlighted the challenges in predicting and defining these diseases. The study emphasizes the need for clearer criteria to identify and manage ultra-rare diseases effectively.

The study is significant in advancing the understanding of ultra-rare diseases in pediatric populations of Saudi Arabia, however falls short in scientific rigor and validity of the findings. I have few comments as below.

1. Integrate functional genetic analysis to elucidate the functional significance and pathogenicity of identified genetic variants. Incorporating functional assays or in silico prediction tools enhances the understanding of genotype-phenotype correlations and disease mechanisms.

2. Provide detailed interpretation of genetic variants, including their potential impact on protein function, disease pathogenesis, and clinical relevance.

3. The study reports varying responses to antiseizure medications among patients, but does not thoroughly investigate factors contributing to this variability.

Incorporating these suggestions can enhance the quality and impact of the existing study focusing on ultra-rare diseases in pediatric populations.

6. PLOS authors have the option to publish the peer review history of their article (what does this mean?). If published, this will include your full peer review and any attached files.

Reviewer #1: No

Reviewer #2: **Yes: **Naseem Akhter

---

## [Author Response · Author response to Decision Letter 0]

19 Jun 2024

Journal Requirements:

Author response:

Dear editor, kindly note that the requirements are revised and the submission has been updated accordingly. 

2. For studies involving third-party data, we encourage authors to share any data specific to their analyses that they can legally distribute. PLOS recognizes, however, that authors may be using third-party data they do not have the rights to share. When third-party data cannot be publicly shared, authors must provide all information necessary for interested researchers to apply to gain access to the data. (https://journals.plos.org/plosone/s/data-availability#loc-acceptable-data-access-restrictions) 

Author response:

Dear editor, kindly note that a data availability statement at the last page of the main manuscript file (lines 576-580) and we have filled all the information regarding the availability statement in the online submission system.

Author response:

Dear editor, kindly note that the ethics statement was removed and added only to the methods section. 

Author response:

Dear editor, kindly note that the supporting information has been added.

Reviewer #1: Muthaffar et al. used advancement of next-generation sequencing to detect ultra-rare mutations that are barely encountered by the majority of physicians. Ultra-rare and rare diseases cumulatively acquire a prevalence equivalent to type 2 diabetes with 80% being genetic in origin and more prevalent among high consanguinity communities including Saudi Arabia. SCN1A mutation was reported among 19 patients out of total 30 patients. Other mutations included SZT2, ROGDI, PRF1, ATP1A3, and SHANK3. The results suggest that ultra-rare diseases must be viewed as a distinct category from rare diseases with potential demographic and clinical hallmarks. Additional objective and descriptive criteria to detect such cases are needed.

This is an excellent study that cover new aspect of next generation sequencing technique to detect ultra rare mutations in the population. I have some comments before the acceptance of this manuscript for the publications.

Author response:

Thank you for taking the time to review our manuscript. We appreciate your positive feedback and acknowledgment of the importance of our study on the detection of ultra-rare mutations using next-generation sequencing. Kindly find our response to each query raised in your revision. 

Comments:

1. Introduction of the study is very brief need more details with latest reference from the literature is required.

Author response: 

Dear reviewer, we thank you for your productive feedback the introduction section. We appreciate your suggestion to provide more details and incorporate the latest references from the literature to enhance the overall understanding of our study. We incorporated three, up-to-date and relevant references, from the literature to support our study. The newly added references are (4), (5), and (12) that can be found in lines 62, 65, and 74, respectively. The most recent one is as recent as the year 2024 and others from 2023 and 2022. We would also like to note that all references used in the articles were used while taking into consideration how relevant and updated they are.

2. It would be helpful to provide more specific information on the prevalence of reported mutations in the Saudi population and the impact of consanguineous marriages on the occurrence of genetic diseases.

Author response: 

Dear reviewer, we thank you for your productive feedback. The impact of consanguinity on rare, ultra-rare diseases, and genetic disorders were further elaborated in lines 78-82. However, the detailed data on the prevalence is scarce. Multiple research studies demonstrated a general surge in rare diseases in Saudi Arabia compared to other regions globally and attributed such findings to the high consanguinity (Eissa et al., 2021; Aleissa et al., 2022) that was thoroughly described in the introduction with additional references for your kind interest. 

3. Please provide the ethical approval of the study.

Author response: 

Dear reviewer, we thank you for your productive feedback. Kindly note that the ethical statement was added in the supplementary section, however, we moved it into the methods in the first section as per the journal instructions. You may notice that the statement is detailed and long; and this's due to the sensitive nature of the topic and participants. However, if it requires summarization we would be pleased to do so. The updated segment is in lines 97-106. 

4. Also, add the details either the consent from patients were taken for this study?

Author response: 

Dear reviewer, we thank you for your productive feedback. Kindly note that the ethical statement was added and this query was addressed in the previous query. We added a consent statement that can found in lines 102-105. 

5. It is not clear how many samples were used for WGS and how many samples for WES.

Author response:

Dear reviewer, thank you for your question regarding the number of samples used for WGS and WES. We appreciate your interest in this aspect of our study. However, we would like to clarify that our research primarily focuses on the detection of ultra-rare mutations using NGS techniques, rather than providing an extensive analysis of the sample sizes used for WGS and WES in which other research might discuss its properties and sensitivity to detect mutations. Therefore given the scope and objectives of our study, we have provided the necessary details on the methods and results to support our findings. While sample size is an important consideration in genetic studies, our primary aim was to highlight the prevalence and genetic basis of URD. Consequently, we believe that including specific information on the number of samples used for WGS and WES may not be directly relevant to the main focus of our research.

We hope that through our study, we have provided valuable insights into the detection of ultra-rare mutations and their implications for clinical practice. We appreciate your understanding and consideration of the scope of our research and if further amendments are needed in this area we would happily do so. Also please note that some information were added in result section on the use of WGS/WES according to each mutation (line 206).

6. It would be better to add graphical representation of the data as done in figure 1. Add more figures that will add value to the manuscript.

Author response:

Dear reviewer, we thank you for your valuable comment. Kindly note that two additional graphs have been added in lines 188 to represent epidemiological distribution of the detected cases in figure 1 and in line 189 for a gender-genetic illustration in figure 2. Also, figure 3 was recreated to improve consistency among graphs and enhance its quality. All figure will be attached in the submission in a separate file as per the journal requirements. 

7. Detailed methodology is required such as WES and WGS etc.

Author response:

Dear reviewer, we thank you for your valuable comment. Kindly refer to our answer to your query number 5. We would be pleased to address additional queries in this regard if needed.

8. The results section presents the identified variant in the epileptic genes. However, it would be beneficial to include the frequency or prevalence of this variant in the Saudi population, if available. This information would help assess the rarity or novelty of the variant and its potential contribution to the reported clinical presentation.

Author response:

Dear reviewer, we thank you for your valuable comment. Kindly note that the prevalence of these ultra-rare mutation is scarce in global literature, nevertheless, our local literature. For example among our reported mutation is SCN1A which is becoming increasingly recognized in the literature in comparison to the other mutations. However, we arguably have only a single cohort from Saudi patient conducted by Bashiri et al. 2023 and included only 4 patients (8.9%) out of a total cohort of 45 patients. This prevalence for example cannot be projected as national prevalence. Other mutations are ever more rare and have little to no local literature available. We address this limitation that been made by lack of literature by thoroughly describing our patients and clearly identifying their genetic data such as in lines 200-201, 206-207, 211-212, and other. We hope this satisfies you and we would be pleased to add additional reporting if needed. 

9. What was inclusion and exclusion criteria of the patients added in this study.

Author response:

The inclusion of our patients' included the following: 

- Pediatric patients (line 119)

- Following up at the pediatric neurology clinics (line 119)

- Confirmed diagnosis of an URD using NGS (line 120)

- The diseases classification as an URD was made according to well-established criteria such as NORD classifications (line 120-121)

Furthermore, additional details on the subjects and population were demonstrated in the methodology, study design and setting section (line 108). 

10. The conclusion summarizes the main findings of the study. However, it would be helpful to reiterate the clinical implications of the identified genes and highlight its relevance for genetic counseling and patient management. This would emphasize the practical implications of the research.

Author response:

Thank you for your valuable feedback on our manuscript. We appreciate your suggestion to reiterate the clinical implications of the identified genes and highlight their relevance for genetic counseling and patient management in the conclusion. We completely agree with the importance of emphasizing the practical implications of our research. Therefore, a new segment has been added to the conclusion in lines 400-414.

11. Some grammatical mistakes and missing reference are seen need to be corrected in the revised manuscript.

Author response:

Thank you for reviewing the language of our manuscript. We appreciate your valuable feedback and have carefully considered your comments regarding grammatical mistakes and missing references by our authors, a native speaker, and using language improvement softwares. In the revised version of the manuscript, we have taken the following actions to address these issues: 

1. We have thoroughly proofread the manuscript and corrected all the identified grammatical errors. For example lines 14, 17, 18, 117, 124, 133, and in other places that can be found in the tracked version. 

2. We apologize for the oversight in not including certain references in the original manuscript. We have now carefully reviewed our reference list and cross-referenced it with the citations in the text.

Finally,

I am particularly grateful for your insightful comments that have prompted me to reconsider certain aspects of the research methodology and data analysis. Your suggestions for additional experiments or analysis have been duly noted and will be incorporated into future work. I also appreciate the time and effort you have dedicated to reviewing the manuscript.

 

Reviewer #2: Present retrospective multicenter study focused on pediatric patients with ultra-rare mutations, revealing their prevalence and clinical characteristics. Results highlighted the challenges in predicting and defining these diseases. The study emphasizes the need for clearer criteria to identify and manage ultra-rare diseases effectively.

The study is significant in advancing the understanding of ultra-rare diseases in pediatric populations of Saudi Arabia, however falls short in scientific rigor and validity of the findings. I have few comments as below.

1. Integrate functional genetic analysis to elucidate the functional significance and pathogenicity of identified genetic variants. Incorporating functional assays or in silico prediction tools enhances the understanding of genotype-phenotype correlations and disease mechanisms.

Author response:

Thank you for your valuable feedback on our manuscript. We appreciate your suggestion to integrate functional genetic analysis. We agree that incorporating functional assays or in silico prediction tools can greatly enhance our understanding of genotype-phenotype correlations and disease mechanisms. However, we would like to clarify that in the current study, functional genetic analysis was not available due to resource limitations and the ultra-rare nature of the diseases under investigation. As a result, we were unable to perform experimental functional assays or access reliable in silico prediction tools specific to these ultra-rare genetic variants. Nonetheless, we have made efforts to provide a comprehensive interpretation of the potential impact of the identified genetic variants based on existing literature, known functional domains of the affected genes, and the observed clinical responses to antiseizure medications. We acknowledge that performing functional analysis would have added valuable insights to our study and strengthened our conclusions.

We sincerely appreciate your suggestion, and while functional analysis was not possible in the current study and hence added to the limitation, we will carefully consider your feedback for future research endeavors. A new limitation has been added in lines 394-398. 

2. Provide detailed interpretation of genetic variants, including their potential impact on protein function, disease pathogenesis, and clinical relevance.

Author response:

Dear reviewer, kindly note that in our study, we investigated several genetic mutations in patients, and I will discuss each mutation, including the SCN1A, SZT2, ROGDI, PRF1, ATP1A3, and SHANK3 mutations, providing a detailed interpretation of their potential impact on protein function, disease pathogenesis, and clinical relevance. Please refer to following lines:

- SCN1A (lines 308-313)

- SZT2 (lines 313-319) 

- ROGDI (lines 319-321)

- PRF1 (lines 321-324)

- ATP1A3 (lines 324-330)

- SHANK3 (lines 330-338)

The elaboration was associated with additional references that we cited with taking into consideration how relevant and recent the source is. 

3. The study reports varying responses to antiseizure medications among patients, but does not thoroughly investigate factors contributing to this variability.

Incorporating these suggestions can enhance the quality and impact of the existing study focusing on ultra-rare diseases in pediatric populations.

Author response:

Dear reviewer, kindly note that additional areas of discussion has been added to discuss potential causes of seizure response disparities. We discussed the genetic factors, pharmacokinetics/pharmacodyn

---

## [Decision Letter · Decision Letter 1]

5 Jul 2024

Clinical Description and Evaluation of 30 Pediatric Patients with Ultra Rare Diseases: A Multicenter Study with Real-World Data from Saudi Arabia

PONE-D-24-05215R1

Dear Dr. Anas,

We’re pleased to inform you that your manuscript has been judged scientifically suitable for publication and will be formally accepted for publication once it meets all outstanding technical requirements.

Kind regards,

Laith Al-Eitan

Academic Editor

PLOS ONE

Additional Editor Comments (optional):

Reviewers' comments:

Reviewer's Responses to Questions

**Comments to the Author**

1. If the authors have adequately addressed your comments raised in a previous round of review and you feel that this manuscript is now acceptable for publication, you may indicate that here to bypass the “Comments to the Author” section, enter your conflict of interest statement in the “Confidential to Editor” section, and submit your "Accept" recommendation.

Reviewer #1: All comments have been addressed

Reviewer #2: All comments have been addressed

2. Is the manuscript technically sound, and do the data support the conclusions?

Reviewer #1: Yes

Reviewer #2: Yes

3. Has the statistical analysis been performed appropriately and rigorously? 

Reviewer #1: Yes

Reviewer #2: Yes

4. Have the authors made all data underlying the findings in their manuscript fully available?

Reviewer #1: Yes

Reviewer #2: Yes

5. Is the manuscript presented in an intelligible fashion and written in standard English?

Reviewer #1: Yes

Reviewer #2: Yes

6. Review Comments to the Author

Reviewer #1: (No Response)

Reviewer #2: (No Response)

7. PLOS authors have the option to publish the peer review history of their article (what does this mean?). If published, this will include your full peer review and any attached files.

Reviewer #1: **Yes: **Dr. Muhammad Imran Naseer

Reviewer #2: **Yes: **Naseem Akhter

---

## [Editor Report · Acceptance letter]

9 Jul 2024

PONE-D-24-05215R1 

PLOS ONE

Dear Dr. Alyazidi, 

I'm pleased to inform you that your manuscript has been deemed suitable for publication in PLOS ONE. Congratulations! Your manuscript is now being handed over to our production team.

Kind regards, 

on behalf of

Professor Laith Al-Eitan 

Academic Editor

PLOS ONE